# Gene Network Analysis of Alzheimer’s Disease Based on Network and Statistical Methods

**DOI:** 10.3390/e23101365

**Published:** 2021-10-19

**Authors:** Chen Zhou, Haiyan Guo, Shujuan Cao

**Affiliations:** School of Mathematical Sciences, Tiangong University, Tianjin 300382, China; zhouchen19970803@163.com (C.Z.); guohaiyan2020@163.com (H.G.)

**Keywords:** Alzheimer’s disease, network pharmacology, network entropy, network topology, Bayesian algorithm, logical regression algorithm

## Abstract

Gene network associated with Alzheimer’s disease (AD) is constructed from multiple data sources by considering gene co-expression and other factors. The AD gene network is divided into modules by Cluster one, Markov Clustering (MCL), Community Clustering (Glay) and Molecular Complex Detection (MCODE). Then these division methods are evaluated by network structure entropy, and optimal division method, MCODE. Through functional enrichment analysis, the functional module is identified. Furthermore, we use network topology properties to predict essential genes. In addition, the logical regression algorithm under Bayesian framework is used to predict essential genes of AD. Based on network pharmacology, four kinds of AD’s herb-active compounds-active compound targets network and AD common core network are visualized, then the better herbs and herb compounds of AD are selected through enrichment analysis.

## 1. Introduction

Alzheimer’s disease (AD) is a chronic age-associated neurodegenerative disorder, and there are no definitive treatments or prophylactic agents. Its pathological features include senile plaque, nerve fiber tangles, and massive loss of neurons [1]. As its pathogenesis is not clear, clinical drugs used commonly can only relieve symptoms within a certain period of time but cannot improve the disease fundamentally.

Network pharmacology is associated with drug targets and human disease genes. On the basis of understanding the “drug-target gene-disease gene” network, the effects of different drugs on different target proteins are evaluated by using network analysis methods [2,3].

Many different computational methods have been employed for the different application fields. Gianni D’Angelo and Francesco Palmieri proposed a novel autoencoder-based deep neural network architecture, where multiple autoencoders are embedded with convolutional and recurrent neural networks to elicit relevant knowledge about the relations existing among the basic features (spatial-features) and their evolution over time [4]. Gianni D’Angelo and Francesco Palmieri described the use of Genetic Programming for the diagnosis and modeling of aerospace structural defects. The resulting approach aims at extracting such knowledge by providing a mathematical model of the considered defects, which can be used for recognizing other similar ones [5]. Zhang et al. proposed a Bayesian regression approach to explain similarities of disease phenotypes by using diffusion kernels of one or several protein-protein interaction (PPI) networks [6]. Chen et al. proposed two improved Markov random field (MRF) algorithms, which can automatically assign weights to different data sources, using Gibbs sampling processes [7,8]. Chen et al. proposed a fast and high-performance multiple data integration algorithm [9] for identifying human disease genes, the logistic regression based algorithm is extended to the multiple data integration case, where the parameters (weights) of different data sources can be tuned automatically.

In this paper, AD genes are collected from multiple databases, and the gene network of AD is constructed by considering some factors such as gene co-expression and metabolic relationship. The gene network is divided into modules by Cluster one [10], MCL [11], Glay [11] and MCODE [11,12]. Then these division methods are evaluated by network structure entropy, and the optimal division method, MCODE. Through functional enrichment analysis, the functional modules are identified. Furthermore, essential genes can be predicted by the analysis of network topology characteristics of these functional modules. In addition, the integrated algorithm (logical regression algorithm under Bayesian framework) is used to predict AD’s essential genes. The final predicted essential genes are obtained by analyzing these two results above.

AD is located in the brain, but it is closely associated to the kidneys, liver, heart, spleen, and other viscera, according to traditional Chinese medicine [13,14]. Compound herbs have the characteristics of multi-components and multi-targets. In this study, we screen out the effective herb compounds for the treatment of AD by identifying the essential genes of AD, the herb-active compound-active target genes network, and the common core network of AD [15,16].

## 2. Materials and Methods

### 2.1. Data Preparation

#### Data Sources

Some common herbs for treating AD are KXS (Kaixinsan), DYSYS (Dangguishaoyaosan), YGS (Yigansan) and YQTYT (Yiqitongyutang). The compounds of these four herbs are obtained [17,18] (see Appendix A). Their active targets were obtained from the Traditional Chinese Medicine Systems Pharmacology (TCMSP) Database [19]. The AD-associated genes were collected from the database of National Center for Biotechnology Information (NCBI) database [20], Online Mendelian Inheritance in Man (OMIM) database [21], and Therapeutic Target Database (TTD) [22]. The PPI dataset is derived from the database of IntAct Molecular Interaction Database (IntAct) [23]. The human gene expression profiles are obtained from the Gene Expression Omnibus (GEO) database [24]. The pathway datasets are obtained from the Kyoto Encyclopedia of Genes and Genomes (KEGG) database [25]. The human protein complexes are from the database of Comprehensive Resource of Mammalian protein complexes (CORUM) database [26].

### 2.2. Methods

#### 2.2.1. Prediction of Essential Genes based on Modular Network

##### Network Module Partition Method

According to the distribution of network nodes in the module, the module division method can be divided into overlapping modules and non-overlapping modules. The common algorithms, MCODE, MCL, Glay, and cluster one, are used to divide the network. The first three algorithms are non-overlapping algorithms, while the last one is an overlapping algorithm. In this paper, the above four-module partition methods are used to divide AD networks. 

##### Entropy 

Recently, “Shannon entropy” has been introduced to measure some properties of networks, also known as “network entropy”. Its value can effectively assess the stability of the network. The smaller numerical value of network entropy, the more stable the network [27]. Network structure entropy is used as the evaluation method. Let N and $k_{i}$ denote the number of nodes, the degree of the *i*-th node, respectively. The entropy of a network [28] is defined as follows:(1)$$E=-\overset{N}{\underset{i=1}{\sum }}I_{i}\ln I_{i}\;\;where\;\;\;\;I_{i}=\frac{k_{i}}{\sum_{i=1}^{N}k_{i}}$$

##### Prediction of Functional Gene Modules

The correlation between AD original network and divided module network is discussed based on gene function enrichment analysis and association indices [29,30]. Jaccard association index is often used to evaluate the functional correlation between each module and the original network [29]. In addition, Fuxman Bass Juan et al. survey many association indices, such as Simpson, Geometric, Cosine, PCC (Pearson Correlation Coefficient) [31]. Zhu and Qiao et al. further extend the PCC association index to measure the correlation between each module and the function of the original network [32], as shown in Table 1.

##### Screening of Essential Genes

Research on the essential genes can help us to understand the biology of the disease. Various tools have been developed to predict and judge the essential genes in the network [33]. In this paper, the network topology attributes of functional modules are analyzed by 11 indexes of Cyto-Hubba [33], such as degree centrality (DC), betweenness centrality (BC), closeness centrality (CC), density of maximum neighborhood component (DMNC), maximum neighborhood component (MNC), bottleneck (BN), edge percolated component (EPC), maximum clique centrality (MCC), edge clustering coefficient (ECC), radiality and clustering coefficient.

#### 2.2.2. Integrated Algorithm for Predicting Essential Genes

Chen et al. proposed a fast and high-performance multiple data integration algorithm for identifying human disease genes [9]. The disease gene identification problem was first expressed as a two-classification problem, and the feature vectors of each gene were extracted from the integrated network. Combined with the binary logistic regression model, maximum likelihood estimation and Bayesian idea, the model parameters are estimated, and the posterior probability of each gene was calculated. The final decision score was obtained by calculating the percentage of individual posterior probability.

##### Acquisition of Priori Probability of Genes

Suppose the integrated network contains genes $g_{1}\ldots g_{n+m}$, in which $g_{1}\ldots g_{n}$ are the unknown ones and $g_{n+1}\ldots g_{n+m}$ are the known ones in the OMIM database. Similar to the method used in references [8,9], for i=1…n, if $g_{i}$ belongs to the protein complex, then let its prior probability be:(2)$$P_{i}=\frac{A}{B}$$
where *A* denotes the number of AD genes in the complex and *B* denotes the number of all disease genes in the complex. If $g_{i}$ does not belong to the protein complex, then let its prior probability be:(3)$$P_{i}=\frac{C}{D}$$
where *C* is the number of all known genes of AD and *D* is the total number of human genes.

Then, generate a random number following the standard uniform distribution U(0,1). If the numerical value of the random number is larger than $P_{i}$, then assign 0 as the prior label for $g_{i}$. Otherwise, assign 1 as the prior label for $g_{i}$. The prior probability of AD genes in the OMIM database are ${\hat{P}}_{n+i}=1,i=1\ldots m$.

##### Binary Label Assignment

Assign binary labels according to the prior probability calculated in 2.2.2.1, if ${\hat{P}}_{i}=1,i=1\ldots n+m$, then the binary label is $\hat{x}=1$. If ${\hat{P}}_{i}=0,i=1\ldots n+m$, then the binary label is $\hat{x}=0$.

##### Obtain Feature Vectors according to the Integrated Network and Binary Labels

Only considering direct neighbors to construct feature vectors limits the capability of the method to use other topological attributes in a biological network. Therefore, the number of second order neighbors (indirectly connected) are employed to construct the feature vector [9] as:(4)$$\varphi_{i}={\left(1,\varphi_{i1},\varphi_{i0},\varphi_{i1}^{\prime },\varphi_{i0}^{\prime }\right)}^{T}$$
where, $\varphi_{i1}$ and $\varphi_{i0}$ are the number of direct neighbors of $g_{i}$ that are connected to vertices with labels 1 and 0, $\varphi_{i1}^{\prime }$ and $\varphi_{i0}^{\prime }$ are the numbers of the second-order neighbors of $g_{i}$ that are connected to vertices with labels 1 and 0. All feature vectors of individual genes together form a feature matrix as:(5)$$F_{1}={\left[\begin{array}{ccccc} 1 & \varphi_{11} & \varphi_{10} & \varphi_{11}^{\prime } & \varphi_{10}^{\prime }\\ 1 & \varphi_{21} & \varphi_{20} & \varphi_{21}^{\prime } & \varphi_{20}^{\prime }\\ \vdots & \vdots & \vdots & \vdots & \vdots \\ 1 & \varphi_{N1} & \varphi_{N0} & \varphi_{N1}^{\prime } & \varphi_{N0}^{\prime } \end{array}\right]}_{N\times 5}$$

##### Estimate Parameters and Calculate the Posterior Probability

Given a prior configuration $\hat{X}$ for all vertices, a maximum likelihood estimation (MLE) method can be used to estimate the parameter vector ω.

Parameter vector can be written as:(6)$$\omega ={\left(\omega_{0},\omega_{1},\omega_{2},\omega_{3},\omega_{4}\right)}^{T}.$$

The likelihood function can be written as:(7)$$L\left(\omega ;x_{1},x_{2}\cdots x_{N}\right)=\prod_{i=1}^{N}P(x_{i}|\varphi_{i},f).$$

The logistic sigmoid function can be written as:(8){P(xi=1|φi,f)=ef(φi)ef(φi)+1P(xi=0|φi,f)=1ef(φi)+1.

Among them, the linear function
(9)$$f\left(\varphi_{i}\right)=\omega^{T}\varphi_{i}.$$

The log likelihood function of (7) can be written as:(10)$$\ln L\left(\omega ;x_{1},x_{2}\cdots x_{N}\right)=\sum_{i=1}^{N}\ln P(x_{i}|\varphi_{i},f).$$

From (8) and (10), we get
(11)$$\ln L\left(\omega ;x_{1},x_{2}\cdots x_{N}\right)=\sum_{i=1}^{N}\left[x_{i}\omega^{T}\varphi_{i}-\ln(1+e^{\omega^{T}\varphi_{i}})\right].$$

Then, a unique global optimal solution can be found by solving a convex optimization problem. The parameter vector ω is obtained by calculating the maximum value of (11). Then calculate the posterior probability of each gene from (8) and (9).

##### Get Decision Score

Considering that a gene has a higher decision score than most genes, it is more likely to be associated with the disease. Therefore, the final decision score is obtained by using the percentage value of the posterior probability [9]. The decision score is calculated as follows:(12)qi=|{j|Pi≥Pj}|n
where $P_{i}$ is the posterior probabilities of each gene and $q_{i}$ is the top percentage value of $P_{i}$ among all those posterior probabilities.

## 3. Results and Discussion

### 3.1. Network Construction

#### 3.1.1. Herb-Active Compound-Target Network

In Appendix A, 14 kinds of herb compound targets are described. Figure 1 shows the network of four herb-active compounds-target genes. In each sub-image, from the inside to the outside, there are herbs, active compounds, ingredients of the active compound and associated target genes. These active compounds and their ingredients are represented by the same color. In Figure 1a, the blue circle stands for herb KXS. The red triangle, green triangle, and yellow triangle stand for KXS’ active compounds Poria Cocos(Schw.) Wolf. (PCW), Panax Ginseng C. A. Mey. (PGCAM), Acoritataninowii Rhizoma (AR), respectively. The red hexagon, green hexagon, and yellow hexagon stand for ingredients of PCW, ingredients of PGCAM, ingredients of AR, respectively. Blue diamond stands for target genes associated with these ingredients. In Figure 1b, blue circle stands for herb DGSYS. Red triangle, purple triangle, navy blue triangle, wathet blue triangle, green triangle, yellow triangle stand for DGSYS’ active compounds Chuanxiong Rhizoma (CXR), Paeoniae Radix Alba (PRA), Angelicae Sinensis Radix (ASR), PCW, Alisma Orientale(Sam.) Juz. (AOJ), Atractylodes Macrocephala Koidz. (AMK), respectively. Red hexagon, purple hexagon, navy blue hexagon, wathet blue hexagon, green hexagon, yellow hexagon stand for ingredients of CXR, ingredients of PRA, ingredients of ASR, ingredients of PCW, ingredients of AOJ, and ingredients of AMK, respectively. Blue diamond stands for target genes. In Figure 1C, blue circle stands for herb YGS. Red triangle, purple triangle, navy blue triangle, wathet blue triangle, green triangle, and yellow triangle stands for YGS’ active compounds AMK, CXR, ASR, PCW, Radix Bupleuri (RB), Uncariae Ramulus Cumuncis (URC), respectively. Red hexagon, purple hexagon, navy blue hexagon, wathet blue hexagon, green hexagon, and yellow hexagon stand for ingredients of AMK, ingredients of CXR, ingredients of ASR, ingredients of PCW, ingredients of RB, ingredients of URC, respectively. Blue diamond stand for target genes. In Figure 1d, blue circle stands for herb YQTYT. Red triangle, purple triangle, navy blue triangle, wathet blue triangle, green triangle, yellow triangle, and orange triangle stands for YQTYT’ active compounds Hedysarum Multijugum Maxim. (HMM), CXR, ASR, PGCAM, Radix Salviae (RS), Radix Paeoniae Rubra (RPR), Codonopsis Radix (CR), respectively. Red hexagon, purple hexagon, navy blue hexagon, wathet blue hexagon, green hexagon, yellow hexagon, and orange hexagon stand for ingredients of HMM, ingredients of CXR, ingredients of ASR, ingredients of PGCAM, ingredients of RS, ingredients of RPR, ingredients of CR, respectively. Blue diamond stands for target genes.

#### 3.1.2. AD Gene Network Construction

First, we collect AD-associated genes from NCBI database, OMIM database, and TTD database, and eliminated data duplications. Then 859 AD-associated genes are obtained. A disease gene network was constructed using the STRING database (input the above genes and select Homo sapiens, Figure 2a), which consists of 746 genes and 10,920 edges. In addition, another PPI network is obtained from the IntAct database. Then, an initial integrated network, which includes 4210 genes and 21,664 edges, is generated by merging the above interaction networks.

Similar to the method used in reference [9], considering the expression status of 13,416 human gene products and containing 79 human tissues in the GEO database (GSE1133), the PCC value between genes is calculated. A pair of genes are linked by an edge if the PCC value is larger than 0.5. Therefore, the gene co-expression network is constructed. Then, we select those genes and edges that appeared in two biological networks (an initial integrated network and a gene co-expression network). The information of AD pathway is added to the integrated network. Pathway datasets are obtained from the database of KEGG and another AD network generated based on three mini metabolic networks [34]. A pair of genes are linked by an edge if they co-exist in any pathway or network. Finally, a multi-database integrated network includes 2017 genes, and 85,152 edges is obtained (Figure 2b). In Figure 2, nodes stand for AD-associated genes from multiple databases, edge of a pair of nodes stand for interaction between nodes.

### 3.2. Prediction of Essential Genes based on Modular Network

#### 3.2.1. Module Partition

The integrated network is divided into modules by Cluster one, MCL, Glay and MCODE. The gene network modules under different division methods are obtained, the corresponding network entropy is calculated (Table 2).

The AD network is divided into 18 modules by MCODE method, its network entropy is 6.05, which is the lowest. Therefore, MCODE is the optional division method. The score of each module based on MCODE method is defined as the product of the density of the subgraph and the number of vertices (genes) in the sub-graph (DC×|V|), which reflects the density of each node in the modules [12]. The number of genes and score of each module are shown in Table 3 (ignoring a single gene).

#### 3.2.2. Calculation of Association Indices

In order to explore the correlation between the original network and the divided module network in biological function, KEGG enrichment analysis is carried out on the original AD network and module networks. The final results show that there are 146 pathways involved in the original AD network. These modules, divided by MCODE method, cover 136 pathways with a coverage rate of 93.16%. It shows that these divided modules can express most of the functions of the original AD network.

We count pathway numbers of each module by enrichment analysis, intersection numbers of pathway numbers between original network and each module, union numbers of pathway numbers between original network and each module, the gene proportion of each module in original network, shown in Table 4. Some modules (12, 14 and 18) are enriched to 0 pathways, so they are ignored. Module 1 contains 400 genes, accounting for 20.04% of the total number of genes, and it can be enriched to 132 pathways, 128 of them are consistent with the original network pathways.

Some association indices (Jaccard, Simpson, Geometry, Cosine and PCC), are calculated shown in Figure 3. Module 1 is key module in the AD gene network.

#### 3.2.3. Prediction of Essential Genes 

Essential genes can perform their function to a greater extent than other genes in the disease gene network. Module 1 is most representative in AD division modules by MCODE. We use 11 network ranking indexes in Cyto-Hubba to sort the genes in module 1 and select the top 100 genes in each index. Those genes that appear more than six times in the top 100 genes are selected as essential genes of AD. Table 5 shows the repetition times of the genes in module 1 by 11 algorithms. 

These genes contain many known AD disease genes: APP, ACHE, ADAM10, APOE, CHRM1, CHRM3, PSEN1, PSEN2 and so on. In addition, CHAT, DR6, NFKB, BACE1, IDE, PP2A, GSK3B appear in the metabolic network of AD [34] (Table 5), which shows that module 1 is a key module and can be used to predict essential genes instead of the original network.

### 3.3. Integrated Algorithm for Predicting Essential Genes

The posterior probability of candidate genes in AD disease network are calculated by an integrated algorithm. Table 6 shows the relevant information of the top 30 candidate genes (2017 candidate genes in total).

Table 6 shows that the known AD genes APP, ADAM10, ACHE and APOE are in the prediction results. Further, the receiver operating characteristic (ROC) analysis is employed as the evaluation criteria to confirm the performance advantage of Integrated algorithm by varying a threshold for determining positives. The first positive control genes are those known AD disease genes from the pathway of KEGG (hsa05010: Alzheimer’s disease), the negative control genes are selected from leukemia genes and diabetes genes that do not associate with AD genes in an integrated network. The relationship between the true positive rate (TPR) and the false positive rate (FPR) is shown with a blue line in Figure 4; the area under the ROC curve (AUC) is 0.984. The second positive control genes are those disease genes from the AD network generated based on three mini metabolic networks [34], the negative control genes are selected from leukemia genes and diabetes genes that do not associate with AD genes in an integrated network, the relationship between TPR and FPR is shown with a green line in Figure 4, the AUC is 0.916. These results demonstrate that Integrated algorithm can identify essential genes of AD.

### 3.4. Screening of Essential Genes of AD

The essential genes are obtained by using the modular network algorithm and integrated algorithm. The common genes between the above two algorithms as final essential genes of AD (Table 7).

### 3.5. Herb- Active Compounds-Target Genes-Essential Genes Network

There are many similar genes between the target genes of the herb compound and the essential genes of AD (Table 8), which indicates that herbs may act on compound targets to regulate disease-related proteins indirectly, whereas herbs can act on these AD proteins directly. AD’s herb (KXS, DYSYS, YGS, YQTYT)-active compound-active compound targets-AD gene network and similar genes are visualized (Figure 5).

### 3.6. Enrichment Analysis of Herb Compound Target

The Gene Ontology (GO) enrichment analysis (including Biological Process (BP), Cell Component (CC), Molecular Function (MF)) and KEGG pathway enrichment analysis are described in Appendix A, these similar genes can be enriched into AD-associated pathways, which indicates that these similar genes are significantly associated with a response to AD.

We count the number of similar genes between the target genes of the herb compound and essential genes of AD, the number of GO enrichment analysis items and the number of KEGG pathway enrichment analysis items, as shown in Figure 6.

We can see from Figure 6 that YQTYT achieves the best performance in GO enrichment analysis items and KEGG pathway enrichment analysis items, while the number of similar genes between YQTYT and essential genes of AD is 17, which indicates that YQTYT is the best herb in four kinds of AD herbs, and YQTYT may have a better therapeutic effect on AD. 

Furthermore, we count the number of similar genes between the target genes of YQTYT compound and genes of AD pathway in the KEGG (hsa05010: Alzheimer’s disease), compound HMM has the largest number of similar genes, followed by compound RS (Figure 7), so HMM and RS are both contributive to the treatment of AD.

## 4. Conclusions

Currently, herbs have an effect on some diseases such as AD, nephropathy. Herbs are more systematic and holistic. However, some studies are still applying the traditional research idea, “one drug-one target-one illness”, which ignores the multi-target and multi-component characteristic of herbs. In this paper, the gene network of AD is constructed by considering some factors such as gene co-expression and metabolic relationship. The modular network algorithm, the logical regression algorithm under Bayesian framework and maximum likelihood estimation, which simplify the gene network and find essential genes highly associated with the AD. By using the idea of network pharmacology, YQTYT is the best herb in four kinds of AD herbs, and YQTYT may have a better therapeutic effect on AD. In addition, HMM and RS are selected as the better herb compounds for AD based on gene function enrichment analysis. Which means the herb compounds may play a major role in the treatment of AD.

Therefore, network pharmacology, network science, machine learning and statistical strategy are expected to find multi-target herb and herb components for the treatment of AD. Theoretical knowledge is provided for the follow-up study of herbs in the treatment of AD, and a feasible scheme is provided for the study of “drug-target-disease”. 

## Figures and Tables

**Figure 1 entropy-23-01365-f001:**
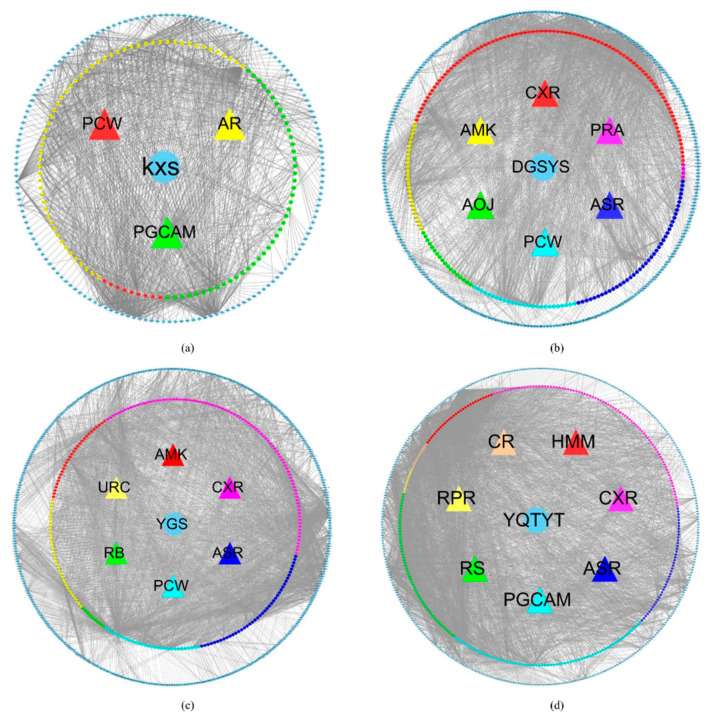
(**a**) KXS-compounds-target genes network. (**b**) DYSYS-compounds-target genes network. (**c**) YGS-compounds-target genes network. (**d**) YQTYT-compounds-target genes network. Blue circle stands for herbs; triangles stand for active compounds; hexagons stand for ingredients; blue diamond stands for target genes.

**Figure 2 entropy-23-01365-f002:**
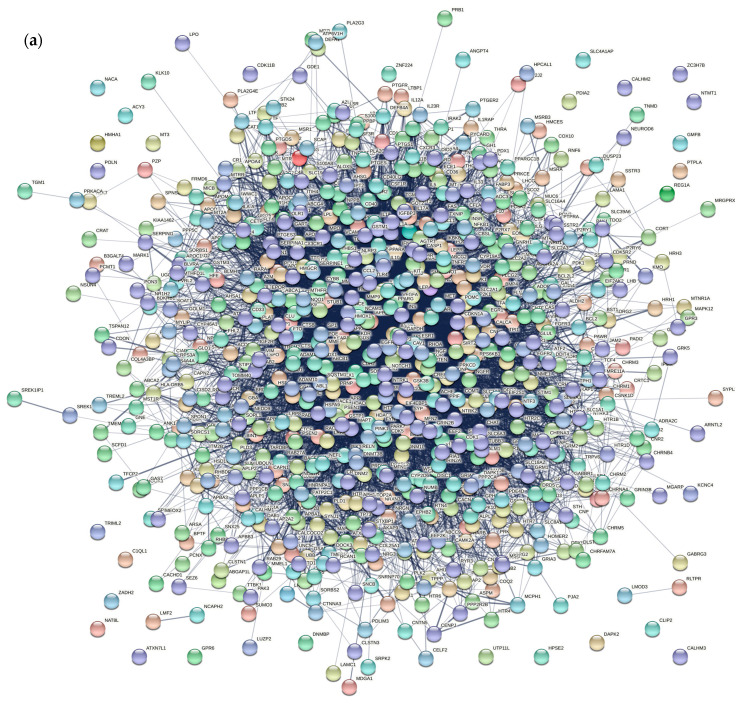
(**a**) The AD disease gene network includes 746 genes and 10,920 edges. (**b**) The integrated network includes 2017 genes and 85,152 edges.

**Figure 3 entropy-23-01365-f003:**
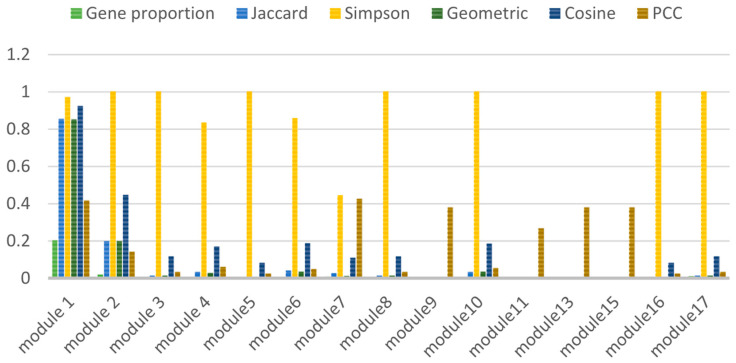
Calculation result of correlation index.

**Figure 4 entropy-23-01365-f004:**
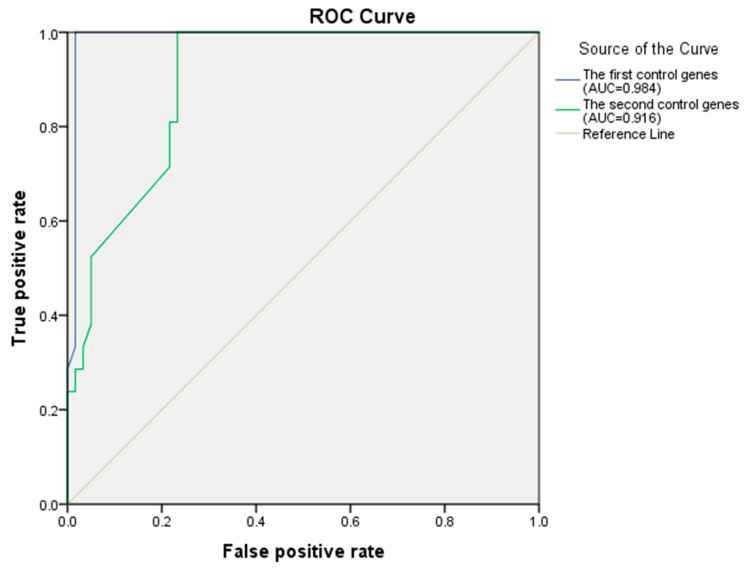
The ROC curve of the Integrated algorithm.

**Figure 5 entropy-23-01365-f005:**
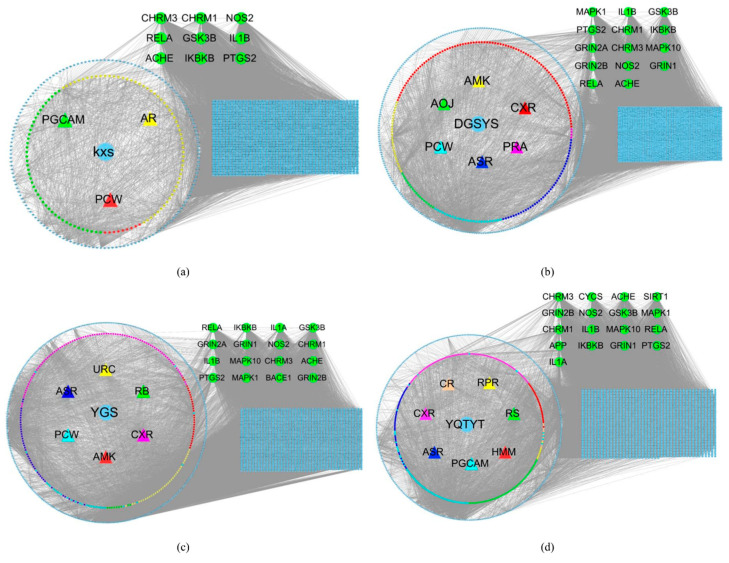
(**a**) KXS-compounds-target genes-AD gene network. (**b**) DYSYS-compounds-target genes-AD gene network. (**c**) YGS-compounds-target genes-AD gene network. (**d**) YQTYT-compounds-target genes-AD gene network. Blue diamond stands for target genes of herb compounds. Green circles stand for similar genes between target genes of herb compounds and essential genes of AD. Blue rectangles stand for genes of AD.

**Figure 6 entropy-23-01365-f006:**
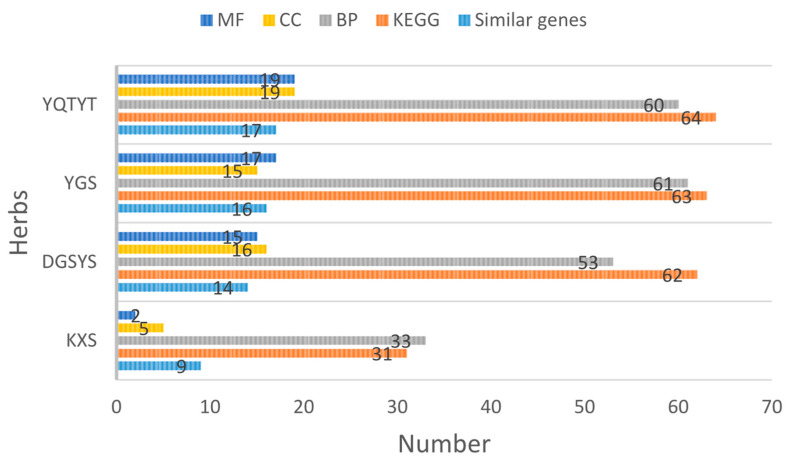
Enrichment analysis results of herb compounds.

**Figure 7 entropy-23-01365-f007:**
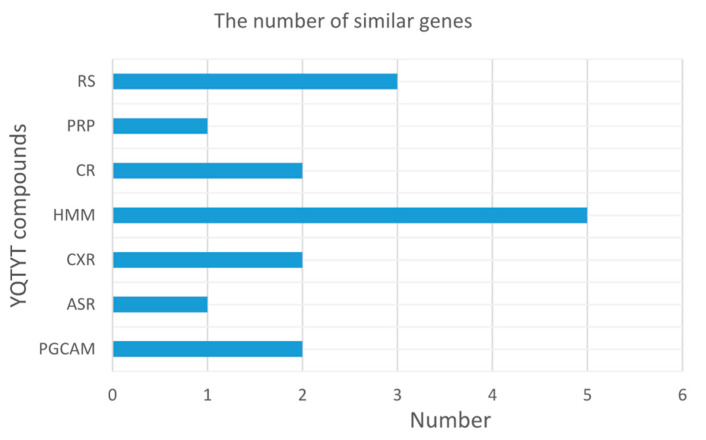
The number of similar genes between target genes of YQTYT compound and genes of AD pathway in the KEGG.

**Table 1 entropy-23-01365-t001:** Correlation index.

Correlation Index	Formula	Meaning
Jaccard	$J_{OC}=\frac{\left|O\;\cap^{\;}C_{i}\right|}{\left|O\cup^{\;}C_{i}\right|}$	The range of values is [0, 1], and the closer it is to 1, the stronger the correlation.
Simpson	$S_{OC}=\frac{\left|O\;\cap^{\;}C_{i}\right|}{\left|min\left(O,C_{i}\right)\right|}$	
Geometric	$G_{OC}=\frac{{\left|O\;\cap^{\;}C_{i}\right|}^{2}}{\left|O\right|\cdot \left|C_{i}\right|}$	
Cosine	$C_{OC}=\frac{\left|O\;\cap^{\;}C_{i}\right|}{\sqrt{\left|O\right|\cdot \left|C_{i}\right|}}$	
PCC	$PCC_{OC}=\frac{\left|O\;\cap^{\;}C_{i}\right|\cdot n-\left|O\right|\cdot \left|C_{i}\right|}{\sqrt{\left|O\right|\cdot \left|C_{i}\right|\cdot \left(n-\left|O\right|\right)\cdot \left(n-\left|C_{i}\right|\right)}}$	

Where, O represents the set of the original network pathways; $C_{i}$ represents the set of the *i*-th module pathways after partition.

**Table 2 entropy-23-01365-t002:** Division results.

Division Methods	Number of Modules	Entropy Value
MCODE	18	6.05
MCL	89	6.19
Glay	17	6.20
Cluster one	89	6.22

**Table 3 entropy-23-01365-t003:** MCODE obtains the division and score of each module.

Module	The Number of Genes	Module Score	Module	The Number of Genes	Module Score
1	400	400.000	10	6	3.200
2	38	6.585	11	3	3.000
3	7	5.667	12	3	3.000
4	5	4.500	13	3	3.000
5	5	4.000	14	3	3.000
6	10	3.778	15	3	3.000
7	15	3.571	16	3	3.000
8	4	3337	17	20	2.947
9	7	3.333	18	4	2.667

**Table 4 entropy-23-01365-t004:** Pathway results of each module.

Module	The Number of Pathways	Intersection	Union	Gene Proportion	Module	The Number of Pathways	Intersection	Union	Gene Proportion
1	132	128	150	20.04%	9	2	0	146	0.35%
2	29	29	146	1.90%	10	5	5	148	0.30%
3	2	2	146	0.35%	11	1	0	146	0.15%
4	6	5	147	0.25%	13	2	0	147	0.15%
5	1	1	146	0.25%	15	2	0	148	0.15%
6	7	6	147	0.50%	16	1	1	146	0.15%
7	9	4	147	0.75%	17	2	2	146	1.00%
8	2	2	151	0.20%					

**Table 5 entropy-23-01365-t005:** Essential genes of AD.

Gene	Repetitions	Gene	Repetitions	Gene	Repetitions	Gene	Repetitions
ABCA1	10	NFKB	10	WNT9A	10	WNT3	9
ACHE	10	NMDAR	10	WNT9B	10	WNT4	9
CASP6	10	PKC	10	XBP1	10	APBB1	8
CHAT	10	PP2A	10	ADAM10	9	APH1B	7
CTFA	10	PRPC	10	APP	9	APOE	7
CYLD	10	PSD95	10	BACE1	9	BECN1	7
DAG1	10	SIRT1	10	CHRM5	9	CALM1	7
DR6	10	SPS	10	GRIN1	9	CAPN2	7
EETS	10	UCHL1	10	IDE	9	CDK5	7
EPHB2	10	UQCRB	10	LRP1	9	CHRM1	7
FYN	10	VLDLR	10	MAPT	9	CHRM3	7
GRIN3A	10	WNT1	10	PSEN1	9	CYCS	7
HPETE	10	WNT3A	10	PSEN2	9	DVL2	7
HSPG	10	WNT5A	10	RELA	9	GNAQ	7
IKKA	10	WNT5B	10	TNF	9	GRIN2A	7
IKKB	10	WNT6	10	WNT10B	9	GRIN2B	7
INSP3R	10	WNT7A	10	WNT11	9	GRM5	7
LDLR	10	WNT7B	10	WNT16	9	GSK3B	7
LILRB2	10	WNT8A	10	WNT2	9	HRAS	7
MAPK	10	WNT8B	10	WNT2B	9	IKBKB	7

**Table 6 entropy-23-01365-t006:** Information of the top 30 candidate genes by integrated algorithm.

Gene	Posterior Probability	Score	Gene	Posterior Probability	Score
APP	0.9998	1	GRIN1	0.9927	0.992481
ADAM10	0.9991	0.999499	CDK5R1	0.9926	0.99198
MAPK1	0.9989	0.998997	CDK5	0.9919	0.991479
MAPT	0.9986	0.998496	MAP2K1	0.9918	0.990977
RELA	0.9956	0.997995	AKT2	0.9915	0.990476
ACHE	0.9955	0.997494	MTOR	0.9915	0.990476
MAPK10	0.9952	0.996992	GRIN2C	0.9913	0.989474
APOE	0.995	0.996491	SIRT1	0.9913	0.989474
KIF5A	0.9945	0.99599	CALM1	0.9912	0.988471
NFKB1	0.9944	0.995489	CACNA1D	0.9911	0.98797
GRIN2A	0.994	0.994987	ITPR1	0.9911	0.98797
GNAQ	0.9937	0.994486	ATP2A2	0.991	0.986967
HRAS	0.9935	0.993985	CASP7	0.991	0.986967
GRIN2B	0.9933	0.993484	DVL1	0.991	0.986967
APBB1	0.9929	0.992982	INS	0.991	0.986967

**Table 7 entropy-23-01365-t007:** Predicted essential genes for AD.

Gene	Gene	Gene	Gene	Gene
ACHE	DVL2	ITPR1	NOX1	WNT11
ADAM10	EPHB2	KLC1	NOX4	WNT16
APBB1	GNAQ	LILRB2	NRAS	WNT2
APH1B	GRIN1	LRP1	PPP3R1	WNT2B
APOE	GRIN2A	MAP2K1	PSEN1	WNT3
APP	GRIN2B	MAP2K2	PTGS2	WNT3A
BACE1	GSK3B	MAPK1	RELA	WNT4
CALM1	HRAS	MAPK10	SIRT1	WNT5A
CDK5	IDE	MAPK3	UCHL1	WNT5B
CHRM1	IKBKB	MAPK9	UQCRB	WNT6
CHRM3	IL1A	MAPT	WNT1	WNT7A
CYCS	IL1B	NOS2	WNT10B	XBP1

**Table 8 entropy-23-01365-t008:** Information of similar genes.

Herb	Similar Genes
KXS	ACHE, CHRM1, CHRM3, GSK3B, IKBKB, IL1B, NOS2, PTGS2, RELA
DGYSY	ACHE, CHRM1, CHRM3, GRIN1, GRIN2A, GRIN2B, GSK3B, IKBKB, IL1B, MAPK1, MAPK10, NOS2, PTGS2, RELA
YGS	ACHE, BACE1, CHRM1, CHRM3, GRIN1, GRIN2A, GRIN2B, GSK3B, IKBKB, IL1A, IL1B, MAPK1, MAPK10, NOS2, PTGS2, RELA
YQTYT	ACHE, APP, CHRM1, CHRM3, CYCS, GRIN1, GRIN2B, GSK3B, IKBKB, IL1A, IL1B, MAPK1, MAPK10, NOS2, PTGS2, RELA, SIRT1,

## Data Availability

The data used to support the findings of this study are available from the corresponding author upon request.

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
