# Peer review of "Gene Network Analysis of Alzheimer’s Disease Based on Network and Statistical Methods"

_entropy, 2021, doi:10.3390/e23101365_

Round 1

Reviewer 1 Report

1) In this article authors have screened out the effective herb compounds for the treatment of AD by identifying the essential genes of AD, herbactive compound-active target genes network and the common core network of AD.

2) The action targets are obtained from 3 different databases, are between batch effects taken into consideration and if so how?

3) What was the criteria of choosing these algorithms MCODE cluster, MCLcluster, Glay, clusterone?

4) Did authors also look at marginal probability along with posterior probabilities of each gene?

5) Figure 2 looks overwhelming, please highlight important clusters in network.

6) The AUC is 0.714, not impressive. Any additional modfications in construction may help in improving this number?

7) Figure 5 is also too much information.

Would appreciate authors comments on above questions. 

Author Response

We appreciate the comments and suggestions from the Reviewer, which help to make the manuscript more accurate and clearer. Below, we include point-by-point replies to the comments of the Reviewer, where all changes have been displayed in red font in the manuscript.

  1. The Reviewer commented:

In this article authors have screened out the effective herb compounds for the treatment of AD by identifying the essential genes of AD, herb active compound-active target genes network and the common core network of AD.

Thanks for the reviewer’s comment.

  1. The Reviewer commented:

The action targets are obtained from 3 different databases, are between batch effects taken into consideration and if so how?

Thanks for the reviewer’s suggestion. We collected AD-associated genes from 3 different databases (NCBI database, OMIM database, and TTD database), and eliminated data duplications. Then 859 AD-associated genes are obtained. A disease gene network is constructed by using the STRING database (input the above genes and select Homo sapiens). Gene expression of the three databases were not used for network construction. Batch effect can be ignored in this paper. Then, according to your comment, we will focus on batch effect in our follow-up work.

  1. The Reviewer commented:

What was the criteria of choosing these algorithms MCODE cluster, MCL cluster, Glay, cluster one?

Thanks for the reviewer’s suggestion. The common algorithms include MCODE cluster, MCL cluster, Glay and cluster one are used to simplify complex network. We use network structure entropy as the evaluation method. The smaller value of network structure entropy, the more stable the network.

  1. The Reviewer commented:

Did authors also look at marginal probability along with posterior probabilities of each gene?

Thanks for the reviewer’s suggestion. The priori probabilities and posterior probabilities of each gene were considered, marginal probability was not involved.

  1. The Reviewer commented:

Figure 2 looks overwhelming, please highlight important clusters in network.

Thanks for the reviewer’s suggestion. Figure 2 highlights important clusters according to the reviewer's requirements.

  1. The Reviewer commented:

The AUC is 0.714, not impressive. Any additional modfications in construction may help in improving this number?

Thanks for the reviewer’s suggestion. According to the requirements of the reviewer, the relevant threshold was changed and the AUC was recalculated.

  1. The Reviewer commented:

Figure 5 is also too much information.

Thanks for the reviewer’s suggestion. According to the requirement of the reviewer, highlights important information, such as herb, active compounds and similar genes between target genes of herb compound and essential genes of AD.

We appreciate your support and encouragement, and we will continue to improve our manuscript until it can be published smoothly.

Reviewer 2 Report

The paper proposes a modular network algorithm and a logical regression algorithm under Bayesian framework to predict essential genes of Alzheimer's Disease (AD). By using the idea of network pharmacology and statistical, the better herb compounds are selected which may play a major role in the treatment of AD.

I consider that it is a very interesting paper and matches well to the journal.

The results are well shown, and convincing.

In my opinion, the article has the potential to give a significant contribution and advancement to the related literature.

Nevertheless, there are some suggestions which call for a major:

  1. English is poor and needs to be improved. Sometime it is difficult to understand what the authors want to say.
  2. The paper needs of a revision to get rid of few complex sentences that hinder readability, and to eradicate typo errors.
  3. The abstract should be rewritten because the idea is not clear.
  4. The introduction is too short. It should include: the context, the problem, the state-of-the-art and the proposal. Also, I suggest to extend the Introduction by reporting the main contributions (novelties) of the proposal.
  5. Besides, for the sake of completeness, the introduction should be enhanced with a very short introduction to the usage and to the effectiveness of the Machine Learning and Neural Network approaches in solving problems as that addressed in this paper. In this regard, the following articles should be mentioned and cited for the different application fields:

"Network traffic classification using deep convolutional recurrent autoencoder neural networks for spatial–temporal features extraction", Journal of Network and Computer Applications, 2021, doi: 10.1016/j.jnca.2020.102890.

"Knowledge elicitation based on genetic programming for non destructive testing of critical aerospace systems", Future Generation Computer Systems, Volume 102, 2020, Pages 633-642, doi: 10.1016/j.future.2019.09.007

  1. The related works is missing.
  2. The conclusions should report more comments regarding the obtained results and future works.

Author Response

We appreciate the comments and suggestions from the Reviewer, which help to make the manuscript more accurate and clearer. Below, we include point-by-point replies to the comments of the Reviewer, where all changes have been displayed in red font in the manuscript.

  1. The Reviewer commented:

English is poor and needs to be improved. Sometime it is difficult to understand what the authors want to say.

Thanks for the reviewer’s suggestion. According to the reviewers' comments, We carefully revised the manuscript.

  1. The Reviewer commented:

The paper needs of a revision to get rid of few complex sentences that hinder readability, and to eradicate typo errors.

Thanks for the reviewer’s suggestion. We have revised the language more clearly and concisely.

  1. The Reviewer commented:

The abstract should be rewritten because the idea is not clear.

Thanks for the reviewer’s suggestion. According to the commented of the reviewer, we have rewritten the abstract.

  1. The Reviewer commented:

The introduction is too short. It should include: the context, the problem, the state-of-the-art and the proposal. Also, I suggest to extend the Introduction by reporting the main contributions (novelties) of the proposal.

Thanks for the reviewer’s suggestion. According to the commented of the reviewer, we have rewritten the introduction, and the Machine Learning and Neural Network approaches are added.

  1. The Reviewer commented:

Besides, for the sake of completeness, the introduction should be enhanced with a very short introduction to the usage and to the effectiveness of the Machine Learning and Neural Network approaches in solving problems as that addressed in this paper. In this regard, the following articles should be mentioned and cited for the different application fields:

"Network traffic classification using deep convolutional recurrent autoencoder neural networks for spatial–temporal features extraction", Journal of Network and Computer Applications, 2021, doi: 10.1016/j.jnca.2020.102890.

"Knowledge elicitation based on genetic programming for non destructive testing of critical aerospace systems", Future Generation Computer Systems, Volume 102, 2020, Pages 633-642, doi: 10.1016/j.future.2019.09.007

Thanks for the reviewer’s suggestion. We have carefully read your suggested articles (doi: 10.1016/j.jnca.2020.102890, doi: 10.1016/j.future.2019.09.007), two articles are very useful and have be cited.

  1. The Reviewer commented:

The related works is missing.

Thanks for the reviewer’s suggestion. We have revised the paper more clearly and concisely.

  1. The Reviewer commented:

The conclusions should report more comments regarding the obtained results and future works.

Thanks for the reviewer’s suggestion. According to the requirement of the reviewer, the conclusions more describes the obtained results and future works.

We appreciate your support and encouragement, and we will continue to improve our manuscript until it can be published smoothly.

Kind regards

Round 2

Reviewer 2 Report

The authors have successfully addressed any my question.

Author Response

Dear Editor,

Thank you very much for giving us the opportunity to revise our manuscript again. We appreciate the editor very much for their constructive suggestions on our manuscript entitled "Gene Network Analysis of Alzheimer's Disease Based on Network and Statistical Methods" (ID: entropy-1384317).

We have studied editor' suggestions carefully. According to the editor' detailed suggestions, we have made a careful revision on our manuscript. We provide a point-to-point responses to suggestions raised by editor in following.

Response to Editor

We appreciate the suggestions from the editor, which help to make the manuscript more accurate and clearer. Below, we include point-by-point replies to the suggestions of the editor, where all changes have been displayed in red font in the manuscript.

  1. The Editor suggested:

Many acronyms and mathematical symbols are not clearly defined, which reduces readability. For example the abstract, has many acronyms without a previous definition. The Editor recommends including a list of acronyms and a list of mathematical symbols

Thanks for the Editor’s suggestion. We added a list of acronyms at the end of the paper according to the Editor’s suggestion.

  1. The Editor suggested:

Several figure captions are not sufficiently informative. For example, in fig 4 the caption "ROC Curve" just says what is in the fig title. ON the other hand, the caption of fig 1 seems excessively large. May be to move something to text?

Thanks for the Editor’s suggestion. Captions of Figure 1 and Figure 4 have revised according to the Editor’s suggestion.

  1. The Editor suggested:

Details about the programming for the plots of figs 1, 2 etc should be provided in the text

Thanks for the Editor’s suggestion. Details about the programming for the plots of Figure 1 and Figure 2 have provided in the text according to the Editor’s suggestion.

  1. The Editor suggested:

Check carefully text. The Editor noticed several missing spaces

Thanks for the Editor’s suggestion. We carefully revised the manuscript and have revised the paper more clearly.

  1. The Editor suggested:

figs 6 and 7 needs label in the x-axis.

Thanks for the Editor’s suggestion. We added label in the x-axis of Figure 6 and Figure 7 according to the Editor’s suggestion.

We appreciate your support and encouragement, and we will continue to improve our manuscript until it can be published smoothly.

Kind regards

Shujuan Cao
